# Contribution of Different Subbands of ECG in Sleep Apnea Detection Evaluated Using Filter Bank Decomposition and a Convolutional Neural Network

**DOI:** 10.3390/s22020510

**Published:** 2022-01-10

**Authors:** Cheng-Yu Yeh, Hung-Yu Chang, Jiy-Yao Hu, Chun-Cheng Lin

**Affiliations:** 1Department of Electrical Engineering, National Chin-Yi University of Technology, Taichung 411, Taiwan; cy.yeh@ncut.edu.tw (C.-Y.Y.); geminipig19970530@gmail.com (J.-Y.H.); 2Heart Center, Cheng Hsin General Hospital, Taipei 112, Taiwan; amadeus0814@yahoo.com.tw; 3Faculty of Medicine, School of Medicine, National Yang Ming Chiao Tung University, Taipei 112, Taiwan

**Keywords:** obstructive sleep apnea, single-lead electrocardiogram, filter bank decomposition, convolutional neural network

## Abstract

A variety of feature extraction and classification approaches have been proposed using electrocardiogram (ECG) and ECG-derived signals for improving the performance of detecting apnea events and diagnosing patients with obstructive sleep apnea (OSA). The purpose of this study is to further evaluate whether the reduction of lower frequency P and T waves can increase the accuracy of the detection of apnea events. This study proposed filter bank decomposition to decompose the ECG signal into 15 subband signals, and a one-dimensional (1D) convolutional neural network (CNN) model independently cooperating with each subband to extract and classify the features of the given subband signal. One-minute ECG signals obtained from the MIT PhysioNet Apnea-ECG database were used to train the CNN models and test the accuracy of detecting apnea events for different subbands. The results show that the use of the newly selected subject-independent datasets can avoid the overestimation of the accuracy of the apnea event detection and can test the difference in the accuracy of different subbands. The frequency band of 31.25–37.5 Hz can achieve 100% per-recording accuracy with 85.8% per-minute accuracy using the newly selected subject-independent datasets and is recommended as a promising subband of ECG signals that can cooperate with the proposed 1D CNN model for the diagnosis of OSA.

## 1. Introduction

Obstructive sleep apnea (OSA) is characterized by repeated collapse of the upper airway during sleep. It blocks the airway and then causes shallow and laborious breathing [1]. OSA is very common in patients with cardiovascular disease and is associated with an increased incidence of stroke, heart failure, atrial fibrillation, and coronary heart disease. Severe OSA is further associated with increased all-cause and cardiovascular mortality [2]. OSA affects approximately 9–24% of the general population, but the number of patients who have been diagnosed is very limited, and about 90% of sufferers are still undiagnosed [3]. Hence, early diagnosis and treatment of OSA can reduce adverse human health conditions.

The standard approach for the diagnosis of OSA is based on the respiratory signals (including nasal airflow, thoracic and abdominal movements) and blood oxygen concentration measured by polysomnography. The measurements of respiratory irregularities during sleep include apneas and hypopneas. An apnea is a complete or almost complete cessation of airflow, lasting ≥10 s, and is usually associated with oxygen desaturation. A hypopnea is a reduction in airflow (<70% of a baseline level) associated with oxygen desaturation. The apnea-hypopnea index (AHI) [4], defined as the sum of apneas and hypopneas per hour of sleep, is widely used for diagnosing the severity of OSA, and includes normal (AHI < 5), mild (5 ≤ AHI < 15), moderate (15 ≤ AHI < 30), and severe (AHI ≥ 30) levels. The most serious limitation of polysomnography is that it is inconvenient, time-consuming and expensive. It is an overnight test at a sleep center or hospital and requires numerous electrodes and sensors to monitor various sleep physiological signals.

In recent years, many studies have focused on the development of a more convenient and less expensive OSA diagnostic system based on the analysis of the single-lead ECG signals. Most of them extract and classify the features from the ECG signals, RR intervals, heart rate variability (HRV), or ECG-derived respiration (EDR) signals. It has been shown that the EDR signal can be used to approximate the respiratory rate, and even the respiratory wave morphology [5,6,7]. Hayano et al. [8] further reported that OSA would cause cyclic variation in the heart rate. Hassan et al. [9] extracted the features of the ECG signals based on the tunable-Q factor wavelet transform, and classified the data using a machine learning algorithm, namely random under sampling boosting (RUSBoost). Rachim et al. [10] decomposed the ECG signals into five levels using wavelet decomposition and then extracted 15 features from the detail coefficients (D3–D5). The principal component analysis and support vector machine were applied for feature dimension reduction and classification, respectively. Sharma et al. [11] and Sharma et al. [12] extracted the features from the ECG signals based on the optimal biorthogonal antisymmetric and orthogonal wavelet filter banks, respectively, and introduced the least squares and Gaussian support vector machines (SVM) for classification, respectively. Our previous study [13] proposed a one-dimensional (1D) convolutional neural network (CNN) model which can automatically learn the features of the ECG signals and classify the normal and apnea events. Wang et al. [14] and Wang et al. [15] proposed a modified LeNet-5 CNN model and a deep residual neural network, respectively, to extract and classify the features from RR intervals. The HRV and EDR signals were decomposed into different modes using the variational mode decomposition proposed by Sharma and Sharma [16], and a K-nearest neighbor classifier was designed for classification. Pinho et al. [17] extracted the features from the HRV and EDR signals based on the time-domain and spectral-domain measures and designed the artificial neural networks (ANN) and SVM for classification.

The above-mentioned research has proposed a variety of methods to extract and classify the features from the ECG and ECG-derived signals. However, when we use the 1D CNN model [13] to automatically learn the features of RR intervals from ECG signals, they are easily affected by low-frequency and large amplitude P and T waves. Hence, this study aims to further evaluate whether the reduction in the low-frequency P and T waves can improve the accuracy of detecting apnea events. This study proposed filter bank decomposition with Butterworth bandpass filters to decompose the ECG signal into 15 subband signals, and a one-dimensional (1D) convolutional neural network (CNN) model independently cooperating with each subband to extract and classify the features of the given subband signal. The original subject-dependent and newly selected subject-independent training and test datasets using 70 ECG recordings from the MIT PhysioNet Apnea-ECG database [18,19] were used in this study to evaluate the contribution of different subbands.

The remainder of this paper is organized as follows. Section 2 describes the training and test datasets of one-minute ECG signals from the MIT PhysioNet Apnea-ECG database and demonstrates the proposed apnea detection system based on the filter bank decomposition and 1D CNN model. Results are given in Section 3. A discussion of the study findings is provided in Section 4. Finally, Section 5 concludes this study.

## 2. Materials and Methods

### 2.1. Materials

All of the ECG signals used in this study were obtained from the MIT PhysioNet Apnea-ECG database [18,19] consisting of a training dataset of 35 recordings numbered from a01 to a20, b01 to b05, and c01 to c10, and a test dataset of 35 recordings numbered from x01 to x35. All ECG recordings were measured by polysomnography. The individual recordings vary in length from 401 to 587 min. Each recording contains a single-lead ECG signal digitized at 100 Hz with 12-bit resolution and a set of reference annotations. Each 1-min ECG signal is annotated as A (apnea event) or N (normal event), where A and N denote the presence or absence of apnea, respectively.

According to the metadata of recordings announced by the database, we compared the age, sex, height, and weight of each recording, and further compared the ECG waveforms for the recordings with the same sex, height, and weight to confirm if they belonged to the same study object. It was found that many ECG recordings in the training and test datasets came from the same study subjects. Table 1 shows the summary of the ECG recordings for each study subject in the database. Each subject includes 1 to 4 recordings. These 70 ECG recordings came from 32 study subjects numbered from p1 to p32, consisting of 25 males aged 46.9 ± 9.9 years with body mass index (BMI) 29.7 ± 7.0 $\mathrm{kg}/m^{2}$, and 7 females aged 32.4 ± 7.0 years with BMI 22.1 ± 3.5 $\mathrm{kg}/m^{2}$. Of the 32 study subjects, 18 are marked by * in Table 1 which denotes that they have ECG recordings in both the training and test datasets. For example, the subject p2 has the recording a02 in the training dataset and the recording x14 in the test dataset. Furthermore, 23 of the 35 ECG recordings (x01 through x35) in the test dataset are marked by + in Table 1 which denotes that the ECG recording in the test dataset corresponds to at least one ECG recording in the training set from the same subject. For example, the recording x07 in the test dataset corresponds to three ECG recordings (a05, a10, and a20) in the training set from the same study subject p5. Hence, more than half of ECG recordings in the training and test dataset are subject-dependent.

In order to allow the proposed CNN model to use ECG signals from different subjects during training and testing, in this study we selected new subject-independent training and test datasets from the original 70 ECG recordings in the database. A total of 35 ECG recordings from 16 study subjects marked by @ in Table 1 were selected into the subject-independent training dataset, and the remaining 35 ECG recordings from 16 study subjects were selected into the subject-independent test dataset. Table 2 and Table 3 list the number of normal and apnea events for the subject-dependent and subject-independent training and test datasets, respectively. The numbers of normal and apnea events in the subject-independent datasets are very close to those in the subject-dependent datasets so as to fairly compare their accuracy in detecting apnea events.

### 2.2. The Proposed Sleep Apnea Detection System Based on the Filter Bank Decomposition and the 1D CNN Model

Figure 1 shows the block diagrams of the proposed sleep apnea detection system including the signal preprocessing and 1D CNN model. The input signal is a 1-min ECG signal with a length of 6000 samples in the training and test datasets. The signal preprocessing includes filter bank decomposition and z-score normalization as shown in Figure 1a. The filter bank decomposition was designed to decompose the input ECG signal with a bandwidth of 50 Hz into 2, 4, and 8 equal-bandwidth subband signals with bandwidths of 25 Hz, 12.5 Hz, and 6.25 Hz using the filter banks including 2, 4, and 8 Butterworth band-pass filters with equal-width frequency subbands, respectively. The Butterworth filter is a popular type of filter with a maximally flat passband. The butter function from the Matlab signal processing toolbox [20] was used to implement the fourth-order Butterworth bandpass filters at a sampling rate of 100 Hz. Figure 2a–c depict the magnitude responses of the filter banks, including 2, 4, and 8 Butterworth bandpass filters, respectively. The low-frequency signals below 0.5 Hz and high-frequency signals above 49.5 Hz were filtered to reduce the baseline drift and high-frequency interference.

The z-score function was used to further normalize the decomposed signals and is defined as follows:(1)$$z=\frac{x\;-\;\mu }{\sigma }$$
where x is the input value from the filtered signal, and μ and σ are the mean and standard deviation of the filtered signal, respectively. The z-score is measured in terms of standard deviations from the mean. A z-score of 1.0 indicates an input value that is one standard deviation from the mean. A positive or negative z-score indicates that the input value is above or below the mean. Figure 3, Figure 4 and Figure 5 show examples of the original ECG and the signals after filtering using the filter banks with 2, 4, and 8 Butterworth bandpass filters and z-score normalization, respectively. It can be observed that the subbands higher than 25 Hz, 12.5 Hz, and 12.5 Hz using the filter banks with 2, 4, and 8 filters remove most of the low-frequency P and T waves and reserve the high-frequency components of the QRS waves.

A total of 15 subband signals were obtained in the signal preprocessing stage. Each subband further cooperated with a CNN model as shown in Figure 1b to extract and classify the features of the given subband signal, and to evaluate its accuracy of detecting apnea events. The input of each 1D CNN model only included one subband signal, not multiple subband signals. Each subband cooperating with a CNN model is an independent system for apnea detection.

Figure 6 demonstrates the block diagram of the 1D CNN model proposed in our previous study [13] for feature extraction and classification. There are 10 identical feature extraction layers designed for extracting features from the given subband signal. Each feature extraction layer consisted of a 1D CNN layer with 45 feature maps (Conv-45), a batch normalization layer, an activation layer using the rectified linear unit (ReLU) function, a 1D max pooling layer with a pool size of 2, and a dropout layer with a fraction of 50%. A flattened layer connected after 10 feature extraction layers is used to transform the extracted 2D feature matrix into a 1D feature vector. There are 4 identical classification layers designed for classifying normal and apnea events based on the 1D feature vector. Each classification layer consisted of a fully connected layer with 512 neurons (FC-512), a batch normalization layer, a ReLU activation layer, and a dropout layer with a fraction of 50%. A fully connected layer with 2 neurons (FC-2) connected after 4 classification layers adopted a softmax activation function to calculate the probabilities of the two outputs corresponding to the normal and apnea event, respectively. The classification result is the event with greater probability.

The CNN layers apply the convolution operation to extract the features from the input data. There are 45 filters in the CNN layer to produce 45 feature maps after the convolution operation with the input signal. Each FC layer maps the features from the last layer into the output for final classification. The weights of the CNN and FC layers are initialized by the He normal initialization method [21]. The batch normalization layers are designed to normalize the data to improve the speed, performance, and stability of the proposed CNN model. The max pooling layers are used to reduce the complexity of the network. The use of the pool size of 2 reduces the number of elements in each feature map to one half the size by selecting the maximum element from a pooling window with a 1 × 2 shape. The dropout layers with a dropout rate of 0.5 randomly omit half of the nodes during training to reduce the overfitting that would cause high training accuracy but low test accuracy. The Adam optimizer was applied to train the proposed 1D CNN model to minimize cross entropy [22]. The detailed parameters and output shape can be found in our previous study [13].

## 3. Results

The original subject-dependent and the newly selected subject-independent training and test datasets from the 70 ECG recordings of the MIT PhysioNet Apnea-ECG database were used to assess the performance of the proposed system for detecting normal and apnea events. The performance parameters for per-minute apnea detection, including accuracy (Acc), sensitivity (Sen), and specificity (Spec), were calculated as follows [23]:(2)$$\;\mathrm{Acc}\;(\%)=\frac{\mathrm{TP}+\mathrm{TN}}{\mathrm{TP}+\mathrm{TN}+\mathrm{FP}+\mathrm{FN}}\;\times \;100\%$$
(3)$$\mathrm{Sen}\;(\%)=\frac{\mathrm{TP}}{\mathrm{TP}+\mathrm{FN}}\;\times \;100\%$$
(4)$$\mathrm{Spec}\;(\%)=\frac{\mathrm{TN}}{\mathrm{TN}+\mathrm{FP}}\;\times \;100\%$$
where TP (true positive) and TN (true negative) are the number of events correctly identified as apnea and normal events, respectively, and FP (false positive) and FN (false negative) are the number of events incorrectly identified as apnea and normal events, respectively.

The proposed 1D CNN model was trained and tested using the preprocessed subband signals of the training and test datasets for a given subband, respectively. Each experiment for training and testing included 50 epochs, and the training and testing accuracies were recorded in each epoch. Because the weights of the CNN and FC layers were initialized with random values, only one experiment may obtain underestimated accuracy of the network. Hence, we repeated the experiment five times and selected the highest test accuracy to determine the per-minute accuracy of each subband. Each ECG recording can be further diagnosed as a non-OSA subject or an OSA patient according to AHI based on the results of per-minute apnea detection. The AHI is defined as the average value of 1-min signals which are identified as apnea events per hour. If the AHI is greater than or equal to 5, the ECG recording is diagnosed as an OSA patient, otherwise it is a non-OSA subject [13,14,16,24,25].

Table 4 lists the summary results of the per-minute and per-recording analysis using the ECG signals in different subbands for the subject-dependent and subject-independent test datasets. The per-minute accuracy using the subband signals with the frequency band from 0.5 Hz to 49.5 Hz without z-score normalization can reach 86.1% in the subject-dependent test dataset but is only 74.4% in the subject-independent test dataset. The use of the z-score normalization slightly increased the per-minute accuracy of the subject-dependent test dataset from 86.1% to 86.7% for the frequency band of 0.5–49.5 Hz, but significantly increased the per-minute accuracy of the subject-independent test dataset from 74.4% to 80.7%.

The per-minute accuracies between different frequency bands do not differ greatly in the subject-dependent test dataset, but there is a bigger difference in the subject-independent test dataset. The difference between the per-minute accuracies of 0.5–25 Hz and 25–49.5 Hz using the filter bank with two filters is only 0.2% (87.3% vs. 87.5%) in the subject-dependent test dataset but reaches 6.0% (80.4% vs. 86.4%) in the subject-independent test dataset. The difference between the minimum and maximum per-minute accuracies using the filter bank with four filters is only 2.0% (85.9% of 12.5–25 Hz vs. 87.9% of 25–37.5 Hz) in the subject-dependent test dataset but reaches 4.8% (81.1% of 0.5–12.5 Hz vs. 85.9% of 25–37.5 Hz) in the subject-independent test dataset. The difference between the minimum and maximum per-minute accuracies using the filter bank with eight filters is only 2.7% (85.9% of 6.25–12.5 Hz vs. 88.6% of 18.75–25 Hz) in the subject-dependent test dataset but reaches 6.4% (79.5% of 0.5–6.25 Hz vs. 85.9% of 31.25–37.5 Hz) in the subject-independent test dataset. The highest per-minute accuracy is 88.6% of 18.75–25 Hz with a specificity of 91.5% and sensitivity of 83.8% in the subject-dependent test dataset and is 86.4% of 25–49.5 Hz with a specificity of 87.7% and sensitivity of 84.3% in the subject-independent test dataset.

A higher per-minute accuracy does not always correspond to a higher per-recording accuracy. The highest per-recording accuracies in the subject-dependent test dataset are 100% of 0.5–12.5 Hz with per-minute accuracy of 87.4%, specificity of 93.1%, and sensitivity of 78.1%, and 100% of 31.25–37.5 Hz with per-minute accuracy of 87.5%, specificity of 90.6%, and sensitivity of 82.4%. The highest per-recording accuracy in the subject-independent test dataset is 100% of 31.25–37.5 Hz with per-minute accuracy of 85.8%, specificity of 89.4%, and sensitivity of 80.1%.

## 4. Discussion

The most obvious effect of OSA on ECG signals is the heart rate or RR interval. A previous study reported that OSA would cause cyclic variation of heart rate (CVHR) consisting of bradycardia during apnea followed by abrupt tachycardia on its cessation [8]. In other words, the RR intervals would increase during apnea events, and would decrease after these events. However, when we use the 1D CNN model to automatically extract the features of RR intervals from ECG signals, they are easily affected by low-frequency and large-amplitude P and T waves. Accordingly, in this study it was assumed that if we can reduce the P and T waves to enhance the high-frequency R wave, it would be easier to highlight the characteristics of the RR interval and then improve the accuracy of the proposed apnea detection system.

In order to evaluate whether the reduction of lower frequency P and T waves can increase the accuracy of the detection of apnea events, this study proposed the use of filter banks with two, four, and eight Butterworth bandpass filters to decompose the 1-min ECG signal with a bandwidth of 50 Hz into two, four, and eight equal-bandwidth subband signals with bandwidths of 25 Hz, 12.5 Hz, and 6.25 Hz, respectively. A total of 15 subbands were included in this study. Each subband independently cooperated with a 1D CNN model to extract and classify the features of the given subband signal for evaluating its accuracy of apnea detection. The original subject-dependent and newly selected subject-independent training and test datasets from 70 ECG recordings of the MIT PhysioNet Apnea-ECG database were used to evaluate the accuracies of detecting apnea events for ECG signals in different frequency subbands.

The previous studies proposed various apnea detection methods based on features extracted from ECG and ECG-derived signals. The ECG recordings from the MIT PhysioNet Apnea-ECG database were most commonly used to train and test their proposed methods. Table 5 compares the method and performance of the proposed 1D CNN model with the previous studies for the per-minute apnea detection using subject-dependent datasets from the MIT PhysioNet Apnea-ECG database. The per-minute accuracy of 88.6% using the subband of 18.75–25 Hz proposed by this study outperforms several previous studies [13,14,16,24,25,26] listed in the first part of Table 5 using the same subject-dependent datasets (the original 35 ECG recordings for training and 35 ECG recordings for testing) as this study. This study and the studies of Chang et al. [13], Wang et al. [14], and Li et al. [25] proposed feature-learning-based methods which can automatically learn the features of ECG signals or RR intervals using neural networks. The proposed 1D CNN model only used filtered and normalized 1D ECG signals as input signals and hence did not require additional signal transformation, R-peaks detection, RR interval or EDR calculation. The per-minute accuracy could reach 87.9% in the study of Chang et al. [13]. They used Butterworth bandpass filtering with a preselected frequency band from 0.5 Hz to 15 Hz and z-score normalization for the preprocessing of ECG signals, and the 1-D CNN model for feature extraction and classification. In comparison with this study, they did not evaluate the contribution of different subbands, and only used the original subject-dependent datasets. Wang et al. [14] reported a per-minute accuracy of 87.6%. They proposed a modified LeNet-5 convolutional neural network to automatically extract and classify the features of the input RR intervals. Li et al. [25] achieved 84.7% accuracy for the per-minute apnea detection. They introduced a sparse auto-encoder to automatically extract features and proposed a decision fusion method to improve the classification accuracy. The studies of Sharma and Sharma [16], Song et al. [24] and Surrel et al. [26] focused on feature-engineering-based methods. Sharma and Sharma [16] achieved an accuracy of 87.5% for per-minute classification. They decomposed the HRV and EDR signals into different modes using the variational mode decomposition and used the K-nearest neighbor classifier. Song et al. [24] reported a per-minute accuracy of 86.2% using a sleep apnea detection approach based on the hidden Markov model. Surrel et al. [26] computed apnea-scores for RR intervals and RS amplitudes using a time-domain filtering and power estimation, and classified normal and apnea events using SVM, which can achieve a per-minute accuracy of 85.7%.

The second part of Table 5 further compares several studies which only used the original 35 ECG recordings of the training dataset from the MIT PhysioNet Apnea-ECG database to train and test their models based on the k-fold cross-validation method. Because these studies did not specify that the ECG signals from the same study subject were not distributed across different folds, they would appear in both the training and test datasets, and hence their datasets were also subject-dependent. The accuracy reported by Wang et al. [15] was 94.3%. They proposed a deep residual network to automatically learn the features from the RR intervals and to classify normal and apnea events using the 10-fold cross-validation strategy. The studies of Sharma et al. [11], Sharma et al. [12], and Pinho et al. [17] developed feature-engineering-based methods. Sharma et al. [11] and Sharma et al. [12] reported average classification accuracies of 90.1% and 90.87%, respectively. Both of them extracted features based on the wavelet filter bank and classified normal and OSA groups using SVM. Pinho et al. [17] obtained an accuracy of 82.12%. They selected 20 features from the RR intervals and EDR signals and used the artificial neural network for classification with the 10-fold cross-evaluation method. The study of Surrel et al. [26] listed in the third part of Table 5 further grouped the recordings by subject according to the metadata of recordings including the reported age, sex, height and weight. They reported a patient-specific accuracy of 88%, which used the first ECG recording from each patient to train the SVM classifier, and the other recordings to test it. Hence, their datasets were subject-dependent. Although our performance cannot be directly compared with those of the previous studies listed in the second and third parts of Table 5 due to the use of different methods and datasets, it is worth noting that most of the previous studies adopted the subject-dependent datasets from the MIT PhysioNet Apnea-ECG database.

The main problem with using subject-dependent datasets is that similar ECG signals from the same subject appeared in both the training and test datasets, which may cause accuracy overestimation. Our study results using the original subject-dependent datasets in Table 4 demonstrate that the per-minute accuracies are as high as from 85.9% to 88.6%, and have a high degree of consistency, such that the difference between the minimum and maximum per-minute accuracies is only 2.7%. Hence, the use of the original subject-dependent datasets cannot test the difference in the accuracy of different subbands. This result is different from what we expected above. We expected that the filtered ECG signals with a higher frequency band could better highlight the features of RR intervals and would have a higher accuracy in the detection of apnea events. The possible reason for the highly consistent accuracies may come from the fact that 23 of the 35 ECG recordings (x01 through x35) in the test dataset correspond to at least one ECG recording in the training set from the same subject. That is, the proposed CNN model uses many similar signals from the test dataset during training. Hence, the use of the original subject-dependent datasets may overestimate the accuracy of each subband. This important issue has not been paid attention to by most previous studies.

In order to allow the proposed CNN model to use ECG signals from different subjects during training and testing, this study further selected new subject-independent training and test datasets to train and test the proposed CNN model. It is obvious that the results of the newly selected subject-independent datasets shown in Table 4 can meet our expectations, and they can demonstrate the difference in the accuracy of different subbands. The per-minute accuracy of 86.4% of the higher frequency band of 25–49.5 Hz is 6.0% higher than the 80.4% accuracy of the lower frequency band of 0.5–25 Hz using the filter bank with two filters. The mid-high frequency band of 25–37.5 Hz has the highest per-minute accuracy of 85.9% among the accuracies using the filter bank with four filters, which is 4.8% higher than the 81.1% accuracy of the lowest frequency band of 0.5–12.5 Hz. The per-minute accuracy 85.9% of the mid-high frequency band of 25–31.5 Hz is the highest among the accuracies using the filter bank with eight filters, which is 6.4% higher than the 79.5% accuracy of the lowest frequency band of 0.5–6.25 Hz. Furthermore, the mid-high frequency subbands of 25–49.5 Hz, 25–37.5 Hz, and 25–31.5 Hz improve the per-minute accuracies by 5.7% (86.4% vs. 80.7%), 5.2% (85.9% vs. 80.7%), and 5.2% (85.9% vs. 80.7%), respectively, in comparison with the full frequency band of 0.5–49.5 Hz. Hence, a mid-high frequency band that removes the low-amplitude P and T waves does indeed improve per-minute accuracy of detecting the apnea events in comparison with a low frequency band or a full frequency band.

Table 6 compares the method and performance of the proposed 1D CNN model with the study of Surrel et al. [24] for the per-minute apnea detection using the subject-independent datasets from the MIT PhysioNet Apnea-ECG database. Although both studies used subject-independent datasets, their methods of selecting datasets were different from ours. To the best of our knowledge, only the study of Surrel et al. [24] among the previous studies reported the subject-independent method to train and test their apnea detection system. Their training and test method was similar to the 28-fold cross-validation method, but the ECG signals from the same study subject were not distributed across different folds. They tested the accuracy of 28 patients one by one. The ECG recordings of one of 28 patients were used as the test dataset each time, and 35 recordings selected from the other 27 patients were adopted as the training dataset. Their per-minute accuracy reached 84% using the subject-independent method for training and testing, which is slightly lower than the accuracy of 86.4% reported by this study using the frequency band of 25–49.5 Hz.

If we further compare the results of the subject-dependent and subject-independent methods in the study of Surrel et al. [26], we can find that their subject-independent accuracy of 84% in Table 6 is lower than the subject-dependent accuracies of 85.7% and 88% in Table 5. This result is consistent with this study. Our results in Table 4 show that the per-minute accuracies of the newly selected subject-independent test dataset for all subbands in this study were all lower than those of the original subject-dependent test dataset. The differences are more obvious in the low-frequency subband. For example, 80.4% vs. 87.3% in the subband of 0.5–25 Hz, 81.1% vs. 87.4% in the subband of 0.5–12.5 Hz, and 79.5% vs. 86.4% in the subband of 0.5–6.25 Hz. These results can confirm that the use of the original subject-dependent datasets did overestimate the per-minute accuracy. Hence, the use of the newly selected subject-independent datasets is recommended to train and test the apnea detection system so as to avoid accuracy overestimation, instead of using the original subject-dependent datasets in the MIT PhysioNet Apnea-ECG database.

Although the per-minute accuracy of the subject-independent test dataset in this study can achieve 86.4% using the frequency band of 25–49.5 Hz, the corresponding per-recording accuracy is only 91.4%, with one non-OSA subject and two OSA patients being misdiagnosed. If we consider having better per-minute and per-recording accuracies at the same time, the use of the mid-high frequency band of 31.25–37.5 Hz has a slightly lower per-minute accuracy of 85.8%, but it can reach the per-recording accuracy of 100%.

The main limitation of this study is that the MIT PhysioNet Apnea-ECG database is a relatively small database that only contains 70 ECG recordings. This may affect the generalizability of the study results in clinical applications. Although our results have successfully demonstrated the contribution of different subbands, further investigation with larger clinical populations is required to optimize the proposed apnea detection system.

## 5. Conclusions

The main contribution of this study is that it proposes filter bank decomposition to decompose the ECG signal into 15 subband signals, and a 1D CNN model independently cooperated with each subband to evaluate the accuracies of different subbands for the detection of apnea events. The proposed 1D CNN model was trained and tested using the original subject-dependent and newly selected subject-independent training and test datasets obtained from the MIT PhysioNet Apnea-ECG database. The results indicate that the original subject-dependent test dataset had a highly consistent per-minute accuracy for all subbands and hence could not test the difference in the accuracy of different subbands. Furthermore, the original subject-dependent test dataset overestimated the per-minute accuracy in comparison with the results of the subject-independent test dataset. Hence, the use of the newly selected subject-independent datasets is recommended to train and test the apnea detection system so as to avoid accuracy overestimation. Moreover, the results of the newly selected subject-independent datasets successfully demonstrate that a mid-high frequency band can improve the per-minute accuracy in comparison with a low frequency band or a full frequency band. The use of the frequency band of 31.25–37.5 Hz can reach 100% per-recording accuracy with 85.8% per-minute accuracy using the newly selected subject-independent test dataset and is recommended as a promising subband of ECG signals cooperating with the proposed 1D CNN model for the diagnosis of OSA. The proposed system using the frequency band of 31.25–37.5 Hz and the 1D CNN model can serve as a convenient and advanced diagnosis OSA system. If the AHI estimated by the average value of 1-min apnea events per hour is greater than or equal to 5, it is recommended to follow up with a polysomnography test to confirm the severity of the OSA.

## Figures and Tables

**Figure 1 sensors-22-00510-f001:**
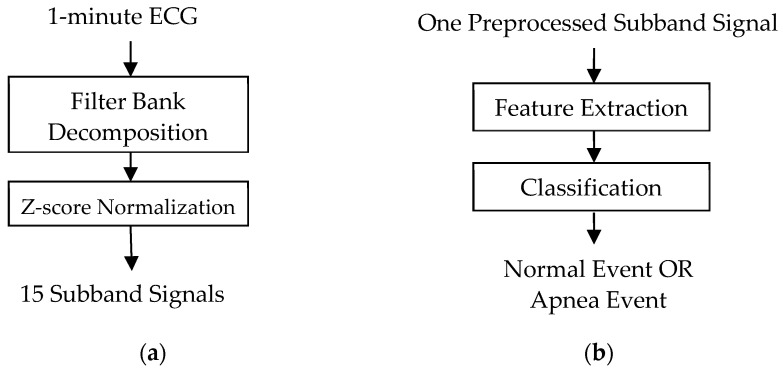
Block diagrams of the proposed sleep apnea detection system including (**a**) the signal preprocessing and (**b**) the 1D CNN model.

**Figure 2 sensors-22-00510-f002:**
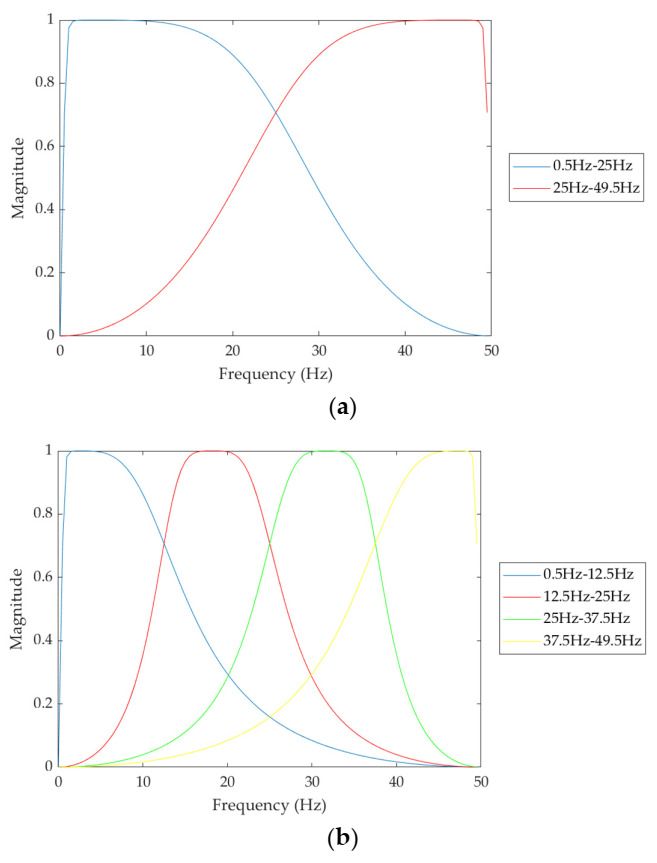
Magnitude responses of the filter banks including (**a**) 2, (**b**) 4, and (**c**) 8 Butterworth bandpass filters.

**Figure 3 sensors-22-00510-f003:**
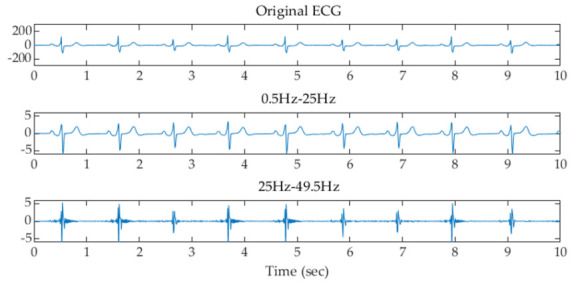
Examples of the original ECG and the signals after filtering using the filter bank with 2 Butterworth filters and z-score normalization.

**Figure 4 sensors-22-00510-f004:**
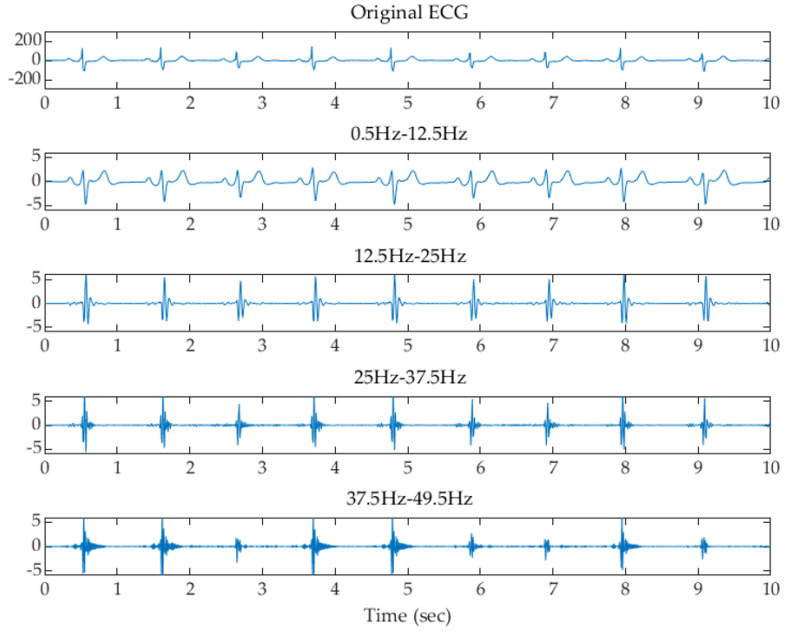
Examples of the original ECG and the signals after filtering using the filter bank with 4 Butterworth filters and z-score normalization.

**Figure 5 sensors-22-00510-f005:**
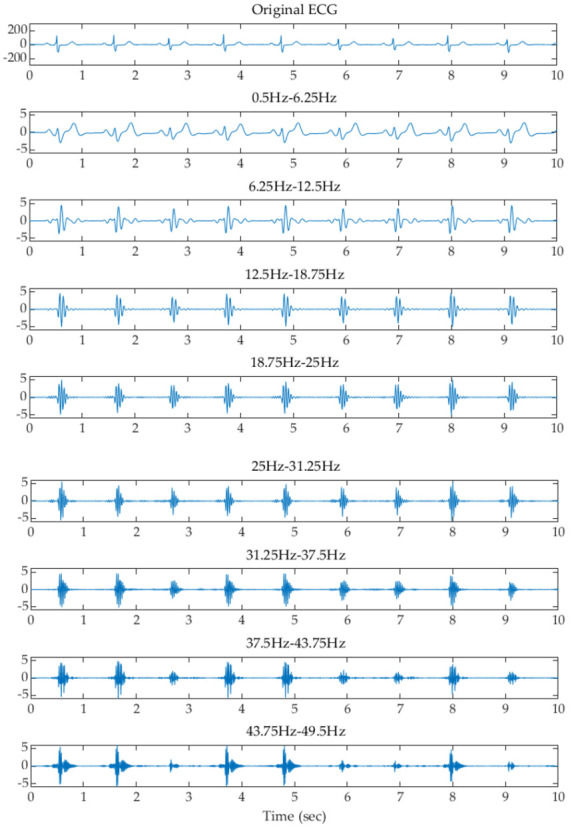
Examples of the original ECG and the signals after filtering using the filter bank with 8 Butterworth filters and z-score normalization.

**Figure 6 sensors-22-00510-f006:**
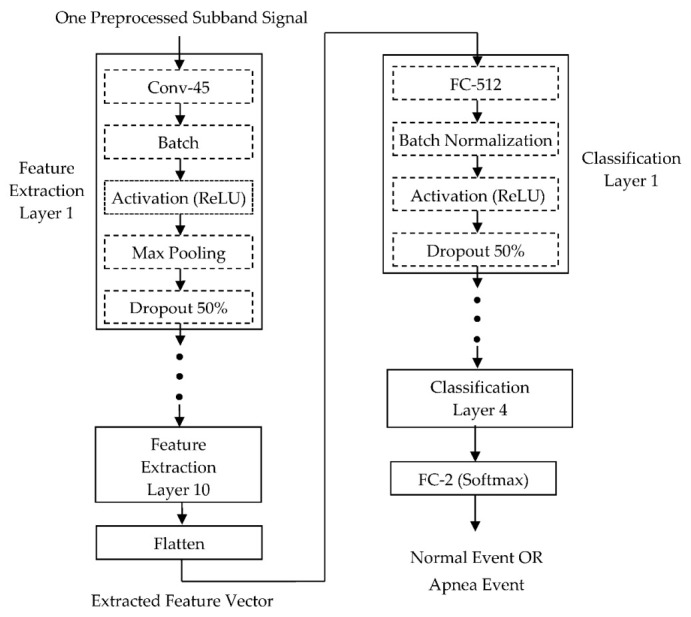
Block diagram of the 1D deep CNN model for identifying normal and apnea events.

**Table 1 sensors-22-00510-t001:** Summary of the ECG recordings for each study subject in the MIT PhysioNet Apnea-ECG database.

Subject No.	ECG Recording No.				Subject No.	ECG Recording No.	
@p1	a01	a14			@p17 *	b05	x11 ^+^
p2 *	a02	x14 ^+^			p18 *	c01	x35 ^+^
@p3 *	a03	x19 ^+^			@p19	c02	c09
@p4	a04	a12			p20 *	c03	x04 ^+^
p5 *	a05	a10	a20	x07 ^+^	@p21 *	c04	x29 ^+^
@p6 *	a06	x15^+^			p22 *	c05	x33 ^+^
@p7 *	a07	a16	x01 ^+^	x30 ^+^	@p23	c06	
p8 *	a08	a13	x20 ^+^		@p24 *	c07	x34 ^+^
p9	a09	a18			@p25 *	c10	x18 ^+^
@p10	a11				p26	x02	
@p11 *	a15	x27 ^+^	x28 ^+^		@p27	x06	x24
@p12 *	a17	x12 ^+^			@p28	x09	x23
p13 *	a19	x05 ^+^	x08 ^+^	x25 ^+^	P29	x10	
@p14 *	b01	x03 ^+^			p30	x13	x26
p15^*^	b02	b03	x16 ^+^	x21 ^+^	p31	x17	x22
p16	b04	c08			p32	x31	x32

* denotes that the study subject has ECG recordings appearing in both the training and test datasets. ^+^ denotes that the ECG recording in the test dataset corresponds to at least one ECG recording in the training dataset from the same subject. @ denotes that all ECG recordings of the study subject are selected into the subject-independent training dataset.

**Table 2 sensors-22-00510-t002:** Number of normal and apnea events for the subject-dependent training and test datasets.

Dataset	No. of Normal Events	No. of Apnea Events	Total
Training	10,512	6511	17,023
Test	10,736	6520	17,256

**Table 3 sensors-22-00510-t003:** Number of normal and apnea events for the subject-independent training and test datasets.

Dataset	No. of Normal Events	No. of Apnea Events	Total
Training	10,662	6350	17,012
Test	10,586	6681	17,267

**Table 4 sensors-22-00510-t004:** Summary results of the per-minute and per-recording analysis using the ECG signals in different subbands for the subject-dependent and subject-independent test datasets.

Frequency Band	Performance Parameters (%) of Per-Minute and (Per-Recording) for the Subject-Dependent Test Dataset			Performance Parameters (%) of Per-Minute and (Per-Recording) for the Subject-Independent Test Dataset		
Using a filter bank with 1 filter but no z-score normalization						
	Acc	Spec	Sen	Acc	Spec	Sen
0.5–49.5 Hz	86.1 (82.9)	89.7 (58.3)	80.1 (95.7)	74.4 (80.0)	91.0 (100.0)	48.2 (72.0)
Using a filter bank with 1 filter and z-score normalization						
	Acc	Spec	Sen	Acc	Spec	Sen
0.5–49.5 Hz	86.7 (94.3)	89.8 (100.0)	81.7 (91.3)	80.7 (82.9)	93.9 (100.0)	59.7 (76.0)
Using a filter bank with 2 filters and z-score normalization						
	Acc	Spec	Sen	Acc	Spec	Sen
0.5–25 Hz	87.3 (97.1)	90.7 (100.0)	81.8 (95.7)	80.4 (82.9)	90.9 (70.0)	63.8 (88.0)
25–49.5 Hz	87.5 (97.1)	88.6 (91.7)	85.7 (100.0)	86.4 (91.4)	87.7 (90.0)	84.3 (92.0)
Using a filter bank with 4 filters and z-score normalization						
	Acc	Spec	Sen	Acc	Spec	Sen
0.5–12.5 Hz	87.4 (100.0)	93.1 (100.)	78.1 (100.0)	81.1 (77.1)	88.3 (50.0)	69.6 (88.0)
12.5–25 Hz	85.9 (88.6)	90.5 (75.0)	78.2 (95.7)	83.4 (94.3)	90.2 (100.0)	72.4 (92.0)
25–37.5 Hz	87.9 (97.1)	89.2 (91.7)	85.6 (100.0)	85.9 (88.6)	87.2 (80.0)	83.7 (92.0)
37.5–49.5 Hz	87.0 (97.1)	88.7 (91.7)	84.2 (100.0)	83.2 (80.0)	89.5 (70.0)	73.3 (84.3)
Using a filter bank with 8 filters and z-score normalization						
	Acc	Spec	Sen	Acc	Spec	Sen
0.5–6.25 Hz	86.4 (88.6)	90.9 (83.3)	79.0 (91.3)	79.5 (80.0)	91.9 (90.0)	59.8 (76.0)
6.25–12.5 Hz	85.9 (94.3)	91.2 (91.7)	77.2 (95.7)	80.3 (94.3)	85.8 (80.0)	71.6 (100.0)
12.5–18.75 Hz	86.3 (94.3)	90.0 (83.3)	80.1 (100.0)	83.9 (91.4)	89.6 (100.0)	74.9 (88.0)
18.75–25 Hz	88.6 (94.3)	91.5 (83.3)	83.8 (100.0)	83.5 (82.9)	88.2 (60.0)	76.1 (92.0)
25–31.25 Hz	88.4 (97.1)	90.2 (91.7)	85.5 (100.0)	85.9 (94.3)	90.2 (100.0)	79.0 (92.0)
31.25–37.5 Hz	87.5 (100.0)	90.6 (100.0)	82.4 (100.0)	85.8 (100.0)	89.4 (100.0)	80.1 (100.0)
37.5–43.75 Hz	87.0 (94.3)	89.4 (83.3)	83.1 (100.0)	82.7 (82.9)	87.5 (70.0)	75.2 (88.0)
43.75–49.5 Hz	87.0 (97.1)	90.3 (91.7)	81.6 (100.0)	82.6 (88.6)	90.5 (90.0)	70.2 (88.0)

**Table 5 sensors-22-00510-t005:** Comparison of the method and performance of the proposed 1D CNN model with the previous studies for the per-minute apnea detection using subject-dependent datasets from the MIT PhysioNet Apnea-ECG database.

Reference	Methods	Subject-Dependent Datasets	Acc (%)
This Study	ECG (18.75–25 Hz Subband) + 1D CNN	The original 35 ECG recordings for training and the original 35 ECG recordings for testing	88.6
Chang et al. [13]	ECG (0.5–15 Hz Subband) + 1D CNN		87.9
Wang et al. [14]	RR Intervals + LeNet-5 CNN		87.6
Li et al. [25]	RR Intervals + Auto-encoder + Decision Fusion		84.7
Sharma and Sharma [16]	HRV + EDR + Feature Engineering + K-nearest Neighbor Classifier		87.5
Song et al. [24]	RR Intervals + EDR + Feature Engineering + HMM-SVM		86.2
Surrel et al. [26]	RR Intervals + RS Amplitudes + Feature Engineering + SVM		85.7
Sharma et al. [11]	ECG + Feature Engineering + LS-SVM	The original 35 ECG recordings for training and testing using 35-fold cross-validation	90.1
Sharma et al. [12]	ECG + Feature Engineering +SVM	The original 35 ECG recordings for training and testing using 35-fold cross-validation	90.87
Wang et al. [15]	RR Intervals + Residual Network	The original 35 ECG recordings for training and testing using 10-fold cross-validation	94.3
Pinho et al. [17]	HRV + EDR + Feature Engineering + ANN	The original 35 ECG recordings for training and testing using 10-fold cross-validation	82.12
Surrel et al. [26]	RR Intervals + RS Amplitudes + Feature Engineering + SVM	Selected 28 ECG recordings for training and selected 43 ECG recordings for testing	88

**Table 6 sensors-22-00510-t006:** Comparison of the method and performance of the proposed 1D CNN model with the previous study for the per-minute apnea detection using subject-independent datasets from MIT PhysioNet Apnea-ECG database.

Reference	Methods	Subject-Independent Datasets	Acc (%)
This Study	ECG (25–49.5 Hz Subband) + 1D CNN	Selected 35 ECG recordings for training and selected 35 ECG recordings for testing	86.4
Surrel et al. [26]	RR Intervals + RS Amplitudes + Feature Engineering + SVM	Selected 35 ECG recordings for training and testing using 28-fold cross-validation	84

## Data Availability

Not applicable.

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
