# Peer review of "Contribution of Different Subbands of ECG in Sleep Apnea Detection Evaluated Using Filter Bank Decomposition and a Convolutional Neural Network"

_sensors, 2022, doi:10.3390/s22020510_

Round 1

Reviewer 1 Report

1-The authors didn’t mention about the quality of the signals. Did you perform any signal quality assessment prior to the signal analysis? Please, provide a table to show the amount of usable and unusable data in this study.

2-Did you perform any algorithms to remove the ECG signal noise and artifacts? Please, describe how the authors solved the noise and artifact issue.

3-There is no information about the devices, which were used for ECG signal recording from the subjects. The authors need to provide the information about the ECG data recording device (if it is available).

4-The authors need to provide the number of 1-min data segments, used for training the model and dependent/independent data testing.

5-How did the authors optimize the structure of the CNN model and hyperparameters of the model?

6-Several previous studies have been mentioned in the discussion and their results have been compared to the present study. However, there are no direct comparisons between this study and the previous studies, in this manuscript. In a fair condition, the authors need to implement the previous methods on the same data (same number of subjects and data time duration), they used in this study, to have a reasonable comparison between the results and then, discuss the advantages and disadvantages of the proposed method compared to the other approaches.

7-Discussion is quite poor. The main hypotheses of the study have not been clearly defined. The research questions and hypotheses need to be described stronger and clearer.

8-Novelty and contribution to knowledge of this research is unclear, which is the main drawback of this study. It is strongly recommended to work on the novelty and contribution of your work as a scientific study.

9-The English language of the manuscript needs to be improved.

Reviewer 2 Report

In this study, the authors aim to detect the Obstructive Sleep Apnea (OSA) from ECG signals. Specifically, they propose a pipeline which is mostly used in previous works but they attempt to enhance their procedure by adding filter bank decomposition and some Butterworth bandpass filters ,so as to decompose the original ECG signals to 2, 4 and 8 subbands. After that, they classify the signals via a 1D CNN model. They test the datasets with two ways, the first in subject-dependent while the second in a subject-independent manner. However, I am not sure that I understood the difference between these two ways. On the first case their results did not achieve to outperform the previous works’ classification accuracies. On the second case, they mention that a specific subband gave the highest results, also higher than the literature’s.

The main problems of the presented work are the following:

  1. The methods are tested only on one dataset
  2. Essentially, the authors did not compare their methodology with e.g., others which use specific features
  3. The authors did not mention at the discussion how the authors of  previous works tested their datasets. Did they use the subject-dependent or subject-independent way?
  4. They mention that one band was the prominent. Would this band be the same for other ECG datasets? Can they generalize?
  5. There exist a lot of sentences that require further explanation, e.g., line 170
  6. In the description of the CNN the authors did not provide any useful information except for the names of each component. They should enhance the description by presenting the role of each component?
  7. In the lines 233-235 the authors mention that they repeated the experiments 5 times and - I believe - that they took the highest classification accuracy. Five times are too low and they should also have to take the average value. Could the authors clarify and discuss.

Overall, I think that it is a “rough” study. The paper needs further information and requires a variety of extra experiments and comparisons. Finally, there are a lot of grammatical mistakes and sentences which confuse the reader.

Round 2

Reviewer 1 Report

The authors have made significant changes to their manuscript. However, I still have many concerns about this manuscript.

1. According to Table 4, performance does not show much difference between different frequency bands and this contradicts the authors hypothesis. As the authors mentioned that:

“our study results in Table 4 further demonstrate that the per-minute accuracies are as high as from 85.9% to 88.6%, and have a high degree of consistency such that the difference between the minimum and maximum per-minute accuracies is only 2.7%.”

However, I believe that the conclusion of consistency is not correct, here. Further, the authors mentioned that:

“This result is different from what we expected above. We expected that the filtered ECG signals with a higher frequency band can could better highlight the features of RR intervals and would have a higher accuracy in the detection of apnea events.”

This sentence confirms that the research hypothesis contradicts the results. So, the main question is that why the authors insisted to decompose the signal into different frequency bands, if the results for all frequency bands are very close? What is the reason to avoid using of the original signal, when various subbands do not show significant difference in their performance?

 It seems that the authors need to change their research hypothesis and discuss the significance of their proposed method against their hypothesis.  

2. As it was asked previously, it is expected that the authors implement other methods to compare with their proposed method, directly. At least, a subselection of the methos mentioned in the second part of the Table 5, could be implemented by the authors and then, the results could be compared. Then, the authors may discuss their proposed method against state of the art.

3. Authors mentioned “newly created subject-independent datasets”, however, this dataset is not created by the authors. It is better to say that the independent dataset is selected from the original dataset by the authors.

4. The authors need carefully check for typos and grammar. For example:

Line 373-374: “We expected that the filtered ECG signals with a higher frequency band can could…”

Further, there are several sentences, which are difficult to understand, especially in Discussion. The English language needs to be improved.

Reviewer 2 Report

Thank you for the clarifications and the extensive editing performed.

Author Response

We thank the referee again for the valuable comments.

This manuscript is a resubmission of an earlier submission. The following is a list of the peer review reports and author responses from that submission.

Round 1

Reviewer 1 Report

This article presents a method for detecting apnea events.

The authors present the measures: Accuracy sensitivity and specificity, but only show results of Accuracy. In the paper cited as [23] are used these three and AUC.

The main drawback is that is presented a result of 88.6% of accuracy in the subband 18.75 Hz-25Hz. But, how the best subband will be selected when there are not test data? This result is compared with other results obtained in articles cited (like [23]), but in those experiments only one result is presented (no one result by subband).

Reviewer 2 Report

The paper "Contribution of Different Subbands of ECG in Sleep Apnea Detection Evaluated Using Filter Bank Decomposition and a Convolutional Neural Network" by Cheng-Yu Yeh et al has some issues that must be solved.    1. Page 2, line 85. 70 ECG recordings- is this number big enough to draw significant conclusions? 2. Page 3, line 106. The recordings were performed during the night or during the day?  3. Page 3, materials. Do you have data about mean age, genre, BMI?  4. Materials. Table 2 and 3. Apnea events are per patient or per sets?  5. page 3, chapter 2.2.  one minute ECG signal x 6000 samples. Do yoy have data about the type of apnea? Obstructive? Central? Mixt? Hypopnea? 6. Results. Training data sets. How can you differentiate apnea from hypopnea? 7. Page 9. The signals were recorded during a PSG?  8. Page 11. Discussion. Do you have data about cardiometabolic comorbidities? They might influence the ECG signal. 9. How about the AHI and OSA severity?  10. How you can apply this methodology in the clinical practice?   

Round 2

Reviewer 1 Report

From the article:

“4. Discussion 267

The study proposed the use of filter banks with 2, 4, and 8 Butterworth bandpass 268 filters to decompose the 1-minute ECG signal into multiple equal-width subband signals, 269 and adopted the same 1D CNN model to extract and classify the features of each subband 270 signal for finding the most promising frequency band in the detection of apnea events.”

A work that wants to ‘find the most promising frequency band’ should work in this way:

  1. Split into train and test
  2. Using only train apply whatever method for finding the best subband and generate a model.
  3. Using that subband apply the model to the test and collect results and measures.

Repeat these 3 steps using different train/test partitions (Cross Validation for instance) and aggregate the measures’ values.

Then the result obtained can be compared to previous published results.

Reviewer 2 Report

Thank for your answers.

I think you can add in the paper some comments form the responses you provided for Q 9 -10.

Round 3

Reviewer 1 Report

The change in:

“4. Discussion

The study proposed the use of filter banks with 2, 4, and 8 Butterworth bandpass filters to decompose the 1-minute ECG signal into multiple equal-width subband signals, and adopted the same 1D CNN model to extract and classify the features of each subband signal for evaluating the contribution of different subbands of the ECG signals to the ac- curacy of apnea detection.”

Is coherent with the article, but in ‘Conclusions’ is said:

“The frequency band of 31.25 Hz-37.5Hz can reach 100% per-recording accuracy with 85.8% per-minute accuracy, and would be a promising subband of ECG signals for the diagnosis of OSA.”

And then is a new sentence:

“The proposed system can serve as a convenient and advanced diagnosis system of 383 OSA only using ECG signals. If the AHI estimated by the average value of 1-minute apnea 384 events per hour is greater than or equal to 5, it is recommended to follow up with a PSG 385 test to confirm the severity of the OSA.”

The system can not be used as diagnosis system as it is not indicated how to combine the different predictions of each subband, or if  the frequency band of 31.25 Hz-37.5Hz will be used then a methodology for selecting a subband should be described and tested.

The main drawback is: to experiment with several subbands and then take the best one to compare con other articles’ result. This is not fair, a methodology to get the best suband should be described and tested.